# TEXT SUMMARIZATION WITH ORACLE EXPECTATION

**Yumo Xu & Mirella Lapata**
Institute for Language, Cognition and Computation
School of Informatics, University of Edinburgh
10 Crichton Street, Edinburgh EH8 9AB
`yumo.xu@ed.ac.uk,mlap@inf.ed.ac.uk`

## ABSTRACT

Extractive summarization produces summaries by identifying and concatenating the most important sentences in a document. Since most summarization datasets do not come with *gold* labels indicating whether document sentences are summary-worthy, different labeling algorithms have been proposed to extrapolate *oracle* extracts for model training. In this work, we identify two flaws with the widely used greedy labeling approach: it delivers suboptimal and deterministic oracles. To alleviate both issues, we propose a simple yet effective labeling algorithm that creates soft, expectation-based sentence labels. We define a new learning objective for extractive summarization which incorporates learning signals from multiple oracle summaries and prove it is equivalent to estimating the oracle expectation for each document sentence. Without any architectural modifications, the proposed labeling scheme achieves superior performance on a variety of summarization benchmarks across domains and languages, in both supervised and zero-shot settings.[1]

## 1 INTRODUCTION

Summarization is the process of condensing a source text into a shorter version while preserving its information content. Thanks to neural encoder-decoder models (Bahdanau et al., 2015; Sutskever et al., 2014), Transformer-based architectures (Vaswani et al., 2017), and large-scale pretraining (Liu & Lapata, 2019; Zhang et al., 2020a; Lewis et al., 2020), the past few years have witnessed a huge leap forward in summarization technology. *Abstractive* methods fluently paraphrase the main content of the input, using a vocabulary different from the original document, while *extractive* approaches are less creative — they produce summaries by identifying and subsequently concatenating the most important sentences in a document — but manage to avoid hallucinations, false statements and inconsistencies.

Neural extractive summarization is typically formulated as a sequence labeling problem (Cheng & Lapata, 2016), assuming access to (binary) labels indicating whether a document sentence should be in the summary. In contrast to the plethora of datasets (see Section 5 for examples) available for abstractive summarization (typically thousands of document-abstract pairs), there are no large-scale datasets with *gold* sentence labels for extractive summarization. *Oracle* labels are thus extrapolated from abstracts via heuristics, amongst which greedy search (Nallapati et al., 2017) is the most popular by far (Liu & Lapata, 2019; Xu et al., 2020; Dou et al., 2021; Jia et al., 2022).

In this work we challenge received wisdom and rethink whether greedy search is the best way to create sentence labels for extractive summarization. Specifically, we highlight two flaws with greedy labeling: (1) the search procedure is *suboptimal*, i.e., it does not guarantee a global optimum for the search objective, and (2) greedy oracles are *deterministic*, i.e., they yield a *single* reference extract for any given input by associating sentences in the document to its corresponding abstract.

Perhaps an obvious solution to the suboptimality problem would be to look for oracle summaries following a procedure based on beam search. Although beam search finds better oracles, we empirically observe that summarization models trained on these do not consistently improve over greedy oracles, possibly due to the higher risk of under-fitting (Narayan et al., 2018a) — there are too few positive labels. Moreover, beam search would also create deterministic oracles. A summarization

---

[1]Our code and models can be found at `https://github.com/yumoxu/oreo`.

Table 1: Sentence labels for a CNN/DM article according to different labeling schemes. Only the first 10 document sentences are shown. Greedy and Beam create oracle summaries (i.e., sentences with label 1) with greedy and beam search, respectively. OREO, our labeling algorithm, incorporates information from multiple summary hypotheses shown in the bar chart ($\mathcal{R}$ is the mean of ROUGE-1 and ROUGE-2). OREO assigns high scores ($> 0.5$) to sentences 1 and 4 which contain an important named entity, Jasmine Coleman, and location, Croydon, South East London. In comparison, greedy and beam labeling consider only one oracle summary, and assign zero to sentences 1 or 4, failing to capture that these are informative and should be probably included in the summary.

| ID | Document Sentence | Greedy | Beam | OREO |
|----|-------------------|--------|------|------|
| 1 | Jasmine Coleman, 12, has been found safe and well some 50 miles from her home. | 0 | 1 | 0.568 |
| 2 | A 12-year-old girl who went missing from her family home at 2 AM amid fears she was driven away by an "older man" has been found safe and well. | 1 | 0 | 1.000 |
| 3 | Jasmine Coleman was reported as missing this morning after disappearing from her home in lancing, west Sussex. | 0 | 0 | 0.429 |
| 4 | The child was found this afternoon following a police appeal some 50miles away in Croydon, South East London. | 1 | 0 | 0.778 |
| 5 | Police feared she may have been driven to London by an older man when they launched an appeal for information this morning. | 1 | 1 | 0.459 |
| 6 | The schoolgirl had not been seen since 11:30 PM on Friday night. | 0 | 0 | 0.000 |
| 7 | Sussex police said she may have been talking with someone on Facetime before disappearing at around 2 am. | 0 | 0 | 0.555 |
| 8 | The force launched a public appeal for information on her whereabouts on Saturday morning. | 0 | 0 | 0.000 |
| 9 | In it, she was described as fair with long, blonde hair and as having possibly been wearing black riding trousers and a polo shirt or a paisley pattern dress. | 0 | 0 | 0.000 |
| 10 | On Saturday afternoon the force confirmed she had been found safe and well in Croydon but could not confirm the circumstances under which police located her. | 0 | 0 | 0.000 |

**Reference Summary**

- Jasmine Coleman disappeared from her home at around 2 AM this morning.

- Police believed she may have been driven towards London by an older man.

- She has been found safe and well in Croydon, South East London today.

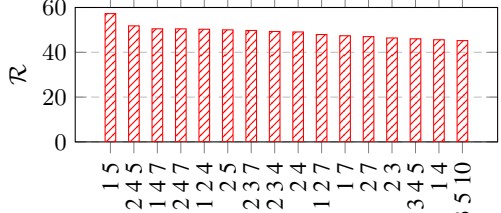

Top-16 Beams (Beam Size $k = 256$)

system trained on either greedy or beam oracles is optimized by maximizing the likelihood of a *single* oracle summary. This ignores the fact that there can be multiple valid summaries for an article, in other words, the summary hypothesis space is naturally a multi-modal probability distribution. We illustrate this point in Table 1.

In this paper we define a new learning objective for extractive summarization which promotes non-deterministic learning in the summary hypothesis space, and introduce OREO, ORacle ExpectatiOn labeling, as a simple yet effective sentence labeling scheme. We prove the equivalence between estimating OREO labels and optimizing the proposed learning objective. As a result, it is sufficient for current models to be trained on OREO labels without requiring any architectural changes.

Extensive experiments on summarization benchmarks show that OREO outperforms comparison labeling schemes in both supervised and zero-shot settings, including cross-domain and cross-lingual tasks. Additionally, we showcase that extracts created by OREO can better guide the learning and inference of a generative system, facilitating the generation of higher-quality abstracts. We further analyze OREO's behavior by measuring *attainable summary knowledge* at inference time, and demonstrate it is superior to related deterministic and soft labeling schemes, which we argue contributes to consistent performance gain across summarization tasks.

## 2 RELATED WORK

Narayan et al. (2018a) were among the first to discuss problematic aspects of sentence labeling schemes for extractive summarization. They argue that labeling sentences individually as in Cheng & Lapata (2016) often generates too many positive labels which leads to overfitting, while a model

trained on greedy labels (Nallapati et al., 2017) underfits the data. Although extractive performance can be boosted via finetuning pretrained encoders with greedy labels (Liu & Lapata, 2019), Zhong et al. (2020) show that reranking summary candidates constructed from greedy predictions can further improve summary quality. This demonstrates that the underfitting problem caused by greedy labels still exists even when pretrained models are used. Issues with greedy labeling have also been discussed from the perspective of *data bias*, including lead bias (Nenkova, 2005; Kedzie et al., 2018; Grenander et al., 2019) — greedy labels display a bias towards lead sentences in news text and systems trained on them do not easily transfer to other domains — and monolingual bias (Jia et al., 2022) — greedy labels created for one language (e.g., English) do not transfer to a different language.

The idea of learning from multiple references has found application in various tasks including dialog response generation (Gao et al., 2019), machine translation (Khayrallah et al., 2020), and question answering (Zhu et al., 2020). Summarization datasets with multiple references are not generally available for model training, but a few have been manually created for system evaluation (Dang, 2005). In extractive summarization, gold references in the form of sentence labels do not usually exist, and learning from multiple references has not been yet explored. In this work, we use beam search to create multiple high-quality *oracle* summaries, from which summary-level supervision is aggregated into sentence labels to promote multi-reference learning for extractive summarization.

## 3 PROBLEM FORMULATION

Let $D = \{x_i\}_1^m$ denote a document consisting of sentences $x_i$. An extractive summarizer produces a *summary hypothesis* that represents salient information via identifying a subset of sentences $\hat{Y} = \{\hat{y}_j\}_1^n, n << m$ within $D$. In practice, ROUGE (Lin & Hovy, 2003), an automatic metric based on lexical overlap is commonly adopted to evaluate $\mathcal{R}(\hat{Y}, S)$, the quality of $\hat{Y}$ against gold *reference summary* $S$.

Following previous work (Cheng & Lapata, 2016; Nallapati et al., 2017; Narayan et al., 2018a; Liu & Lapata, 2019), we conceptualize extractive summarization as a sequence labeling task, and aim to build a system that estimates the summary-worthiness of each sentence in a non-autoregressive manner. As mentioned earlier, sentence labels need to be first extrapolated to train an extractive system, since existing datasets are label-free, they only contain document-abstract pairs. BERTSUM (Liu & Lapata, 2019) is a popular extractive model and representative of the approach sketched above. Built on top of BERT (Devlin et al., 2019), it adds a two-layer Transformer (Vaswani et al., 2017) for sentence representation and a `sigmoid` layer for summary prediction. During inference, document sentences $\{x_i\}_1^m$ are ranked based on their estimated scores, and summary $\{\hat{y}_j\}_1^n$ is identified. The number of $n$ sentences to be included in the summary is often pre-defined and fixed.

## 4 FROM EXISTING LABELING SCHEMES TO OREO

Early labeling methods create sentence labels $\ell_i$ by evaluating the similarity of $x_i$ against reference summary $S$ through various heuristics $h(\cdot)$, $\ell_i \stackrel{\text{def}}{=} h(x_i, S)$, including ROUGE (Lin & Hovy, 2003) and rule-based features such as sentence and paragraph position information, and the number of mentioned entities (Woodsend & Lapata, 2010; Cheng & Lapata, 2016). These methods obtain *local* labels as they assume a sentence can be classified as summary-worthy on its own, without taking the summary context into account.

However, model evaluation does not operate at the sentence-level, as a sentence is part of a summary hypothesis $Y$ together with other candidates (Narayan et al., 2018a). The aim of extractive summarization is to deliver a high-quality summary hypothesis, i.e., a good set of sentences rather than a set of good sentences. A sentence might achieve a high score on its own but contribute little to a summary hypothesis, e.g., due to redundancy. An alternative is to obtain *global* labels, based on whether a sentence occurs within the optimal set of sentences which collectively achieve the highest score according to some evaluation metric like ROUGE:

$$\ell_i \stackrel{\text{def}}{=} \mathbb{1}(x_i \in Y^*) \text{ where } Y^* = \underset{Y \in \mathbb{C}(D)}{\arg\max} \mathcal{R}(Y, S). \tag{1}$$

where $|\mathbb{C}(D)| = C\binom{m}{n}$ is the hypothesis combinatorial space. As Equation (1) is computationally intractable, in practice, it is approximated by further conditioning on a heuristic search space $\mathbb{S}$ such

that $Y^* \approx \arg\max_{Y \in \mathbb{S}(D)} \mathcal{R}(Y, S)$, and the approximated $Y^*$ is usually called an *oracle* summary. A widely adopted approximation is *greedy* labeling (Nallapati et al., 2017; Narayan et al., 2018a; Liu & Lapata, 2019; Zhong et al., 2020), which uses greedy search to maximize $\mathcal{R}$ at each step of sentence selection (the algorithm stops when $\mathcal{R}$ can no longer increase or the maximum number of steps is reached; see Appendix A for the algorithm).

While significantly reducing complexity, greedy labeling does not guarantee a global optimum. To find better summaries to serve as oracles, we propose to replace greedy search with beam search which we refer to as *beam* labeling. We empirically find that around $8\%$–$20\%$ of (greedy) labels can be potentially improved with beam search (when setting the beam size to 256; see Appendix B for details). However, having better labels does not necessarily translate to performance improvements, and we next discuss why this is the case.

## 4.1 OREO: ESTIMATING ORACLE EXPECTATION

Extractive summarization models are typically trained to optimize $\max p_\theta(Y^*|D)$, where the best hypothesis $Y^*$ can be approximated with greedy or beam search. This learning objective maximizes the probability at a single point $Y^*$, and assigns zero probability to other summary hypotheses $\hat{Y}$, regardless of their quality. We note that this formulation leads to a *discrepancy* between how the model is optimized and how the labels against which this optimization takes place are obtained. Given an input document, sequence labeling summarization models assume conditional independence at sentence-level inference, while in greedy labeling, each step in the process of maximizing $\mathcal{R}(Y^*, S)$ conditions on the outcomes of previous steps. From an optimization perspective, this mismatch renders fitting a non-autoregressive sequence labeler difficult for two reasons: (1) learning to search and maximizing the likelihood at $Y^*$ is challenging, and so the model tends to underfit $Y^*$ (Narayan et al., 2018a), and (2) probabilities at *other* $Y$ with high evaluation scores remain *under-estimated* and *uncalibrated* due to supervision sparsity. Simply replacing greedy search with beam search does not resolve these optimization challenges as point estimation is still performed.

A solution is to evaluate summary hypotheses during training and reward the model accordingly (Narayan et al., 2018a). However, this is non-trivial as the the metric $\mathcal{R}$ is usually non-differentiable, and it is computational expensive to sample from a large combinatorial hypothesis space, e.g., with Reinforcement Learning (Sutton & Barto, 1998). Rather than changing the training of the model, in this work, we study how to derive a better sentence labeling algorithm that leads to a better optimization objective.

Specifically, we wish to incorporate multiple high-quality hypotheses as oracle summaries into the learning objective. Our key assumption is that extractive oracles are non-deterministic, but drawn from a distribution $p(Y^*|D, S)$. We thus formulate the objective for extractive summarization as:

$$\max_{Y^* \sim p(Y^*|D,S)} \mathbb{E} \left[ \mathcal{R}(Y^*, S) p_\theta(Y^*|D) \right]. \tag{2}$$

Under this formulation, an optimized model is expected to assign high probability $p_\theta(Y|D)$ when there exists an oracle summary with high probability and high score according to some quality evaluation metric.

From the perspective of sentence labeling, we note that candidate $x_i$ relates to the summarization task through the oracle summary space $\mathbb{Y}$. As $\mathbb{Y}$ is a combinatorial space, the mapping $x_i \to Y^*$ is one-to-many. Therefore, we can compute the probability for each candidate to be selected via marginalization:

$$p(x_i|D, S) = \sum_{Y^*}^{\mathbb{Y}} p(x_i, Y^*|D, S) = \sum_{Y^*}^{\mathbb{Y}} p(x_i|Y^*, D) p(Y^*|D, S). \tag{3}$$

To connect the marginalization in Equation (3) with the summarization objective in Equation (2), we further incorporate hypothesis evaluation $\mathcal{R}(Y^*, S)$, and define the summary-worthiness of a sentence $x_i$ as the expectation of its associated oracle evaluation:

$$\ell_i' \stackrel{\text{def}}{=} \sum_{Y^*}^{\mathbb{Y}} \mathcal{R}(Y^*, S) p(x_i|Y^*, D) p(Y^*|D, S) = \underbrace{\mathbb{E}_{Y^* \sim \underbrace{p(Y^*|D,S)}_{\text{oracle distribution}}}} \left[ \underbrace{\mathcal{R}(Y^*, S)}_{\text{oracle evaluation}} \underbrace{p(x_i|Y^*, D)}_{\text{oracle membership}} \right] \tag{4}$$

---

**Algorithm 1** Labeling with Oracle Expectation

| | |
|---|---|
| 1: **function** OREO($n, k, p$) | 1: **function** STEP($j, \mathcal{B}$)    ▷ Step and beam |
| 2:     ▷ $n$: Max number of sentences in a summary | 2:     Initialize visited paths $\mathcal{V}$ |
| 3:         ▷ $k$: beam size; $p$: oracle distribution | 3:     **for** $b, r \leftarrow \mathcal{B}$ **do** |
| 4:     Initialize beam $\mathcal{B}$ | 4:         **if** $|b| < j$ **then** |
| 5:     **for** $j \leftarrow n$ **do** | 5:             **continue**    ▷ Skip early stopping |
| 6:         $\mathcal{B} \leftarrow$ STEP($j, \mathcal{B}$) | 6:         **for** $i \leftarrow |D|$ **do** |
| 7:     Initialize $\ell'_i$ to $0, \forall i$  ▷ Pre-scaled expectation | 7:             $b' =$ SORT($b + \{i\}$) |
| 8:     **for** $b, r \leftarrow \mathcal{B}$ **do** | 8:             **if** $b'$ not in $\mathcal{V}$ **then** |
| 9:         **for** $i \leftarrow b$ **do** | 9:                 $r' =$ ROUGE($b'$) |
| 10:             $\ell'_i \leftarrow \ell'_i + r * p(b)$ | 10:                 **if** $r' > r$ **then** |
| 11:     $\ell =$ MAXMINSCALE($\ell'$) | 11:                     $\mathcal{B} \leftarrow \mathcal{B} + \{(b', r')\}$ |
| 12:     **return** $\ell$ | 12:             $\mathcal{V} \leftarrow \mathcal{V} + \{b'\}$ |
| 13: **end function** | 13:     **return** TOP-$k(\mathcal{B})$    ▷ Pruned beam |
| | 14: **end function** |

---

where the oracle membership $p(x_i|Y^*, D)$ is identical to $y_i = \mathbb{1}(x_i \in Y^*)$ and the oracle distribution will be discussed in Section 4.3. Given a sequence labeling model $\theta$, maximizing the oracle expectation for all input sentences is equivalent to the objective in Equation (2). We present the proof in Appendix C.

To be compatible with the standard sequence labeling architecture for extractive summarization, we perform MLE with a cross-entropy loss:

$$\min \mathcal{L}(\theta) = \min \sum_{i=1}^{m} \text{CrossEntropy}\left(\ell(x_i), p_\theta(x_i|D, S)\right) \qquad (5)$$

where the scaled expectation $\ell(x_i) = (\ell'_i - \bar{\ell}_{\min})/(\bar{\ell}_{\max} - \bar{\ell}_{\min})$ constitutes the final sentence labels. The details of oracle expectation labeling are given in Algorithm 1.

## 4.2 COMPARISON WITH EXISTING LABELING ALGORITHMS

OREO creates soft (continuous) sentence labels, i.e., it incorporates summary-level evaluation while maintaining low sparsity. A detailed comparison with other labeling algorithms is provided in Table 3. Equation (4) also bears resemblance to the RL objective used in Narayan et al. (2018a): $\max \mathbb{E}_{\hat{Y} \sim p_\theta(Y|D)}[\mathcal{R}(\hat{Y}, S)]$. They evaluate summary hypotheses directly while Equation (4) estimates sentence-level membership. This is a consequence of the nature of the sequence labeler which does not explicitly represent the summary hypothesis space (which is combinatorial), and therefore supervision is delegated to sentences rather than summaries. By maximizing sentence-level likelihood, estimations for the associated hypotheses are updated, albeit indirectly.

Narayan et al. (2018a) employ REINFORCE (Williams, 1992), an on-policy RL algorithm that samples from a model during training, while in Equation (4), samples are drawn from the non-parametrized oracle distribution $p(Y^*|D, S)$. We provide *offline* supervision in the form of static sample labels. In contrast to the *online* reward in RL, offline labels can be reused during the course of training and are therefore more sample efficient. Our labeling scheme can be seen as a type of offline bandit learning (Nguyen-Tang et al., 2022). While the latter has been recently applied to abstractive summarization (Pang & He, 2021), it remains under-explored in extractive summarization.

## 4.3 THE ORACLE DISTRIBUTION

We derive the oracle distribution $p(Y^*|D)$ heuristically bearing in mind that: (a) we have no prior knowledge as to which hypothesis is more or less likely as an oracle summary and therefore assume the oracle distribution to be uniform over a large hypothesis space; and (b) it is desirable for $p(Y^*|D)$ to positively correlate with $\mathcal{R}(Y^*, S)$ and we expect this correlation to become stronger over the course of optimization. In practice, we use beam search (with beam size $k << |\mathbb{Y}|$) to find potential oracle summaries, and adopt a uniform distribution over top-ranked beams: $p(Y^*|D) \sim U(1, t)$, where $t < k$ is a hyper-parameter which we optimize on a development set. We also experimented with several weight annealing mechanisms over top beams as determined by $\mathcal{R}$, as our summary quality evaluation metric (see Appendix D for details).

Table 2: Datasets for monolingual and cross-lingual (last column) summarization. Compression rate denotes the number of sentences extracted to form a summary; and † denotes that trigram blocking (Liu & Lapata, 2019) was applied in sentence selection for redundancy removal.

| Datasets | CNN/DM | XSum | Multi-News | Reddit | WikiHow | MLSum |
|---|---|---|---|---|---|---|
| Language | En | En | En | En | En | En/De/Es/Fr/Ru/Tr |
| Domain | Newswire | Newswire | Newswire | Social Media | Wikipedia | Newswire |
| #Train | 287,084 | 203,028 | 44,972 | 41,675 | 168,126 | 287,227 (En) |
| #Validation | 13,367 | 11,273 | 5,622 | 645 | 6,000 | 13,368 (En) |
| #Test | 11,489 | 11,332 | 5,622 | 645 | 6,000 | 53,981 (Non-En) |
| #Compression Rate | $3^{\dagger}$ | 2 | 9 | 2 | $4^{\dagger}$ | $2^{\dagger}$ |

Table 3: Sentence labeling schemes for extractive summarization. Sum refers to summary-level evaluation. $m$, $n$, and $k$ respectively denote document size, summary size, and beam size.

| Scheme | Sum | Sparsity | Complexity |
|---|---|---|---|
| Local | ✗ | Medium | $\mathcal{O}(m)$ |
| Global | ✓ | High | $\mathcal{O}(\frac{m!}{n!(m-n)!})$ |
| Greedy | ✓ | High | $\mathcal{O}(nm\log m)$ |
| Beam (ours) | ✓ | High | $\mathcal{O}(nmk\log(mk))$ |
| OREO (ours) | ✓ | Low | $\mathcal{O}(nmk\log(mk))$ |

Table 4: Extractive performance (test set, ROUGE-L) on CNN/DM (CD), XSum (XS), Multi-News (MN), Reddit (RD), and WikiHow (WH). We highlight **best** scores and scores *outside the 95% confidence interval* of OREO (using bootstrap resampling; Davison & Hinkley 1997).

| Systems | CD | XS | MN | RD | WH |
|---|---|---|---|---|---|
| LEAD | 36.67 | 14.79 | 38.97 | 14.34 | 23.24 |
| MATCHSUM | 40.38 | 18.41 | 41.89 | 20.13 | 29.58 |
| ORACLE | | | | | |
| Greedy | 48.87 | 23.57 | 44.27 | 28.98 | 32.68 |
| Beam | 52.86 | 23.71 | 46.40 | 29.11 | 36.51 |
| OREO | 50.08 | 20.07 | 46.14 | 24.55 | 34.28 |
| BERTSUM | | | | | |
| Greedy | *39.56* | *17.16* | 41.53 | *19.11* | 28.24 |
| Beam | *39.66* | *17.66* | 41.50 | 19.81 | *25.71* |
| OREO | **39.96** | **17.81** | **41.71** | **20.02** | **28.46** |

# 5 EXPERIMENTS

## 5.1 SUPERVISED EXTRACTIVE SUMMARIZATION

We conducted all extractive summarization experiments with BERTSUM (Liu & Lapata, 2019), the neural summarization architecture introduced in Section 3. We opted for BERTSUM due to its simplicity and popularity in a wide range of summarization tasks (Zhong et al., 2020; Xu & Lapata, 2021; Jia et al., 2022). We nevertheless note that OREO is *model-agnostic* and can be also applied to more complex architectures. For a fair comparison between different labeling schemes, we follow the standard training configuration used in Liu & Lapata (2019) without any additional hyper-parameter optimization (e.g., for our specific labeling scheme). We set $\mathcal{R}$, the summary quality evaluation metric, to the mean of ROUGE-1 and ROUGE-2. We report experiments on a variety of summarization datasets including CNN/DM (Hermann et al., 2015), XSum (Narayan et al., 2018b), Multi-News (Fabbri et al., 2019), Reddit (Kim et al., 2019), and WikiHow (Koupaee & Wang, 2018). Detailed statistics are shown in Table 2.

Our results are presented in Table 4. In the first block, we report the performance of the LEAD baseline which considers the first $k$ sentences in a document as the summary (see last row in Table 4) and MATCHSUM (Zhong et al., 2020), a state-of-the-art system which performs summary reranking with another BERT model. The second block reports ORACLE performance with greedy labels, beam labels ($k = 256$), and OREO labels ($k = 256, t = 16$; we take the top-$n$ sentences with non-zero scores). See Appendix E for the labeling hyperparameters $k, t$ for each dataset, and more detail on experimental settings. The third block reports BERTSUM performance with different labeling schemes.

Although beam ORACLE is superior to greedy ORACLE and raises the upper bound, the overall performance of BERTSUM optimized on beam labels does not significantly improve upon its greedy counterpart. In fact, performance drops drastically on WikiHow. OREO shows inferior ORACLE results as it considers multiple top-ranked beams and is therefore not bound-preserving (see Section 5.5 for detailed analysis). However, BERTSUM trained with OREO labels consistently outperforms a model trained with beam labels, and achieves a 0.18–0.89 ROUGE-L improvement on different

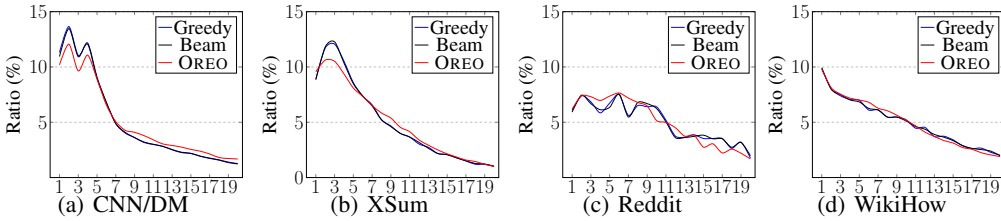

Figure 1: Distribution of label values over sentence positions in documents (development set).

Table 5: Zero-shot cross-domain performance on various test sets for models trained on CNN/DM (ROUGE-L).

| CNN/DM | XS | RD | WH |
|---|---|---|---|
| BERTSUM | | | |
| Greedy | **15.62** | *17.06* | 25.39 |
| Beam | **15.62** | 17.64 | *24.77* |
| OREO | 15.58 | **17.71** | **25.62** |

Table 6: Zero-shot cross-lingual summarization on MLSum (test set, ROUGE-L). Systems with * are supervised. Systems with † use XLM-R large.

| Systems | De | Es | Fr | Ru | Tr | AVG |
|---|---|---|---|---|---|---|
| XLS*† | 41.28 | 21.99 | 24.12 | 10.44 | 33.29 | 26.22 |
| NLS† | 34.95 | 21.20 | 23.59 | 10.13 | 31.49 | 24.27 |
| XLS | | | | | | |
| Greedy | *28.75* | 20.83 | 23.10 | 9.43 | *29.52* | 22.33 |
| Beam | *26.43* | **20.90** | **23.41** | 9.42 | *29.80* | 21.99 |
| OREO | **31.47** | 20.84 | 23.10 | **9.44** | **31.71** | **23.31** |

Table 7: Results for abstractive summarization on CNN/DM (test set). R-1/2/L is a shorthand for ROUGE.

| Systems | R-1 | R-2 | R-L |
|---|---|---|---|
| BART | 44.16 | 21.28 | 40.90 |
| GSUM | | | |
| Greedy | *44.40* | *21.52* | *41.23* |
| Beam | *44.41* | *21.55* | *41.26* |
| OREO | **44.81** | **21.83** | **41.60** |

benchmarks compared to greedy labeling. Although BERTSUM with OREO still falls short of the state-of-the-art, we show that *one-stage* summarization modeling can be enhanced with better labeling, which can serve as a foundation for more complex reranking methods (see Table 11 in Appendix F for details).

## 5.2 ZERO-SHOT CROSS-DOMAIN EXTRACTIVE SUMMARIZATION

We further examine the generalization capability of models trained with OREO labels in a *zero-shot* setting. Specifically, we evaluate a model trained on CNN/DM, against XSum, another news summarization dataset with shorter summaries (at most 2 sentences), and Reddit and WikiHow which represent entirely different domains and topics (discussion forums and instructional text).

Table 5 summarizes our results. Models trained with OREO perform on par with greedy labeling in-domain but display stronger generalization cross-domain. Greedy labels are more prone to lead bias, they deem as summary-worthy sentences mostly from the beginning of the document. Such bias is present in news datasets like CNN/DM but does not transfer to other domains like social media or Wikipedia. OREO alleviates this bias and performs better out-of-domain. As shown in Figures 1(a) and 1(b), OREO is less concentrated on lead sentences in news text.

## 5.3 ZERO-SHOT CROSS-LINGUAL EXTRACTIVE SUMMARIZATION

We next investigate the generalization capabilities of our approach in a cross-lingual setting. We use English data for model training and report *zero-shot* results on a variety of languages from the MLSum dataset (Scialom et al. 2020; see Table 2 for detailed statistics). Following Jia et al. (2022), we augment English articles with word replacement during training (Qin et al., 2021) using the MUSE (Lample et al., 2018) bilingual dictionary to align multilingual representations. We adopt a word replacement rate of 0.5. BERTSUM was initialized with XLM-R base (Conneau et al., 2020), a cross-lingual pretrained model (see XLS in Table 6).[2]

The first block in Table 6, reports the results of a *supervised* XLS model which has access to training data for *all* languages; NLS is the zero-shot state of the art system of Jia et al. (2022); their approach creates multiple sets of greedy labels with different machine translation methods and adopts a neural architecture to learn weights for the obtained label sets. The second block presents the results of

---

[2] We also experimented with mBERT (Devlin et al., 2019) and achieved similar results. See Appendix F.

a zero-shot XLS model with different labeling schemes. As can be seen, OREO labels on Spanish, French, and Russian are on par with greedy labeling. Systems trained with greedy labels exhibit less cross-lingual generalization on German and Turkish, while OREO improves system performance on German by 2.72 ROUGE-L points and on Turkish by 2.19. Previous work (Jia et al., 2022) shows that cross-lingual performance correlates with lead bias in the target language. For example, Turkish articles are less lead-biased than Russian in MLSum, and thus benefit more from better sentence labeling. OREO trails behind NLS which is not surprising as the latter model benefits from more resources, i.e., machine translation and XLM-R large, and a more complex network architecture.

## 5.4 SUPERVISED ABSTRACTIVE SUMMARIZATION

We further assessed whether the proposed labeling scheme is of benefit to abstractive summarization. We experimented with GSUM (Dou et al., 2021), a state-of-the-art abstractive system that takes extractive summaries as additional input to *guide* the generation of document abstracts. During training, GSUM uses extractive oracles as guidance, while at inference time guidance is provided by summary hypotheses produced by a trained extractive system. We initialized GSUM with BART (Lewis et al., 2020), and used BERTSUM as the guidance model optimized with different labeling schemes (i.e., greedy, beam and OREO).

Abstractive summarization results are shown in Table 7. The first block shows the performance of BART (Lewis et al., 2020) which serves as a baseline. In the second block, we report the performance of GSUM (Dou et al., 2021) with greedy labels (default) in addition to beam- and OREO-based variants. As we can see, while beam labeling performs on par with its greedy counterpart, OREO guidance boosts performance with 0.37 ROUGE-L points over vanilla GSUM. We conclude that abstractive systems can also benefit from our expectation-based labeling algorithm, without any modeling changes or hyperparameter optimization. More results with varied guidance settings can be found in Appendix F. Examples of system output are shown in Appendix G.

## 5.5 COMPARISON WITH BOUND-PRESERVING METHODS

Let $\{z_i^*\}_1^m, z_i^* \in \{0, 1\}$ denote a multi-hot representation of an oracle summary. We define a labeling function $\ell$ as *bound-preserving*, if there exists a constant $\gamma \in [0, 1]$ so that the condition $\mathbb{1}\left(\ell(x_i) > \gamma\right) = z_i^*, \forall i$ holds. Intuitively, the condition holds if and only if the top-ranked sentences remain identical. Bound preservation of soft labels guarantees that the performance upper bound of a summarization system trained on soft labels equals that of a system trained on their original hard labels, e.g., obtained via greedy and beam search. For instance, label smoothing (Szegedy et al., 2016), a common technique for training deep neural networks, is bound-preserving. In contrast, OREO is generally *not* bound-preserving for either beam or greedy oracles.[3] To further analyse this property of soft labels, we propose ORMAX as a bound-preserving variant, by replacing the expectation with a max operator: $\ell_i' \stackrel{\text{def}}{=} \max_{Y^* \sim p(Y^*|D, S)} \left[p(x_i|Y^*)\mathcal{R}(Y^*, S)\right]$. Compared to OREO, ORMAX incorporates multiple oracles while additionally preserving the upper bound of beam labels.[4]

Table 8 shows the performance of label smoothing and ORMAX for extractive (first block) and abstractive (second block) summarization. Although label smoothing has been successfully applied to discriminative (Szegedy et al., 2016) and generative NLP tasks (Chen et al., 2018; Lewis et al., 2020), the soft labels it creates do not yield better results than their original hard labels in extractive summarization. Label smoothing performs *implicit model calibration* (Müller et al., 2019), which can potentially improve sentence ranking and selection at inference, however, it also imposes regularization in neural networks training (Szegedy et al., 2016), which may render it less effective for extractive summarization where there is a higher risk of underfitting (Narayan et al., 2018a). On the other hand, ORMAX performs on par with OREO on abstractive summarization, while it underperforms on extractive summarization. Although bound preservation is, intuitively, desirable, our experimental results suggest that it is neither a necessary nor sufficient condition to produce a well-optimized summarization system.

---

[3]Under two special cases our labeling scheme is bound-preserving: (1) with beam size $k = 1$, OREO is equivalent to greedy labeling and (2) with top beam size $t = 1$, OREO is equivalent to beam labeling.

[4]ORMAX is trivially bound-preserving since sentences selected by the top-ranked beam receive the highest score, and the top-ranked beam can be reconstructed by the top-ranked sentences. We illustrate the difference between OREO and bound-preserving methods in Figure 5 in Appendix H.

Table 8: Comparison of OREO to bound-preserving labeling (CNN/DM test set). Results shown for extractive (BERTSUM) and abstractive (GSUM) summarization. LS refers to Label Smoothing ($\alpha$ optimized between $\{0.1, 0.2, 0.3\}$). R-1/2/L is a shorthand for ROUGE.

| Systems | R-1 | R-2 | R-L |
|---|---|---|---|
| BERTSUM | | | |
| Greedy | 43.18 | 20.16 | 39.56 |
| Greedy+LS | ↑0.03 | ↓0.01 | ↑0.03 |
| Beam | 43.25 | 20.14 | 39.66 |
| Beam+LS | ↑0.00 | ↓0.02 | ↑0.01 |
| OREO | 43.58 | 20.43 | 39.96 |
| ORMAX | ↓0.20 | ↓0.27 | ↓0.20 |
| GSUM | | | |
| OREO | 44.81 | 21.83 | 41.60 |
| ORMAX | ↑0.04 | ↓0.02 | ↓0.00 |

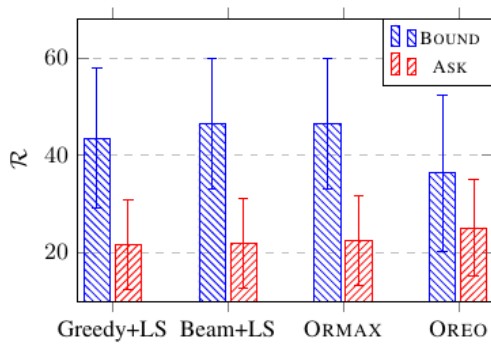

Figure 2: Upper bound and attainable summary knowledge captured by sentence labeling method (CNN/DM validation set) for Label Smoothing (+LS), ORMAX, and OREO.

## 5.6 PERFORMANCE ANALYSIS

Previous experiments revealed that oracles are not necessarily indicative of model performance (see Tables 4 and 8), due to the discrepancy between model optimization and sentence labeling, as discussed in Section 4.1. To further understand how different labeling schemes (and the oracles based on them) influence model performance, we quantify this discrepancy via a sampling-based method which simulates sentence selection for a sequence labeling model at inference.

Bearing in mind that a non-autoregressive sequence labeling model performs conditionally independent predictions and selects a fixed size number of $n$ sentences, we construct summary hypotheses $\hat{Y} = \{\hat{y}_j\}_{j=1}^n$ by drawing *independent* sentence samples, and measure the extent to which a model can Attain Summary relevant Knowledge (ASK) as:

$$\text{ASK} \stackrel{\text{def}}{=} \underset{\{\hat{y}_j\}_{j=1}^n \sim p(x_i|D,S)}{\mathbb{E}} \left[ \mathcal{R}(\{\hat{y}_j\}_{j=1}^n, S) \right] \text{ where } p(x_i|D,S) = \frac{\ell_i}{\sum_{i=1}^m \ell_i} \quad (6)$$

Note that $p(x_i|D,S)$ is shaped by soft labels, and thus results in varied sentence/summary samples for different labeling schemes. The comparison in Figure 2 explains why we observe performance gains from OREO despite obtaining the lowest upper bound performance. The latter considers only the best case scenario at inference, ignoring the fact that some summary knowledge encoded in sentence labels can be hard or impossible to attain, e.g., when sentences in the oracle summary are highly co-dependent (and is therefore challenging to select them jointly with a model making independent predictions), or the oracle summary contains less than $n$ sentences (which again entails that information is missing). Compared to other labeling schemes, OREO captures richer summary information that is attainable for sequence labeling models, narrowing the distance between upper bound performance and ASK. Consistent with our analysis, systems trained on OREO perform robustly on a wide variety of summarization tasks.

## 6 CONCLUSIONS AND FUTURE WORK

We provided a comprehensive analysis of existing labeling schemes for extractive summarization, and identified two flaws in greedy labeling, namely it delivers suboptimal and deterministic labels. We proposed a novel optimization objective to learn from multiple oracle summaries, which can be instantiated by a labeling scheme based on oracle expectation. Experimental results show that the proposed scheme achieves substantial improvement across domains and languages, without any architectural modifications. Our framework is agnostic to the labeling metric $\mathcal{R}$, however, an important future direction is to incorporate different learning signals and provide sentence labels with more desirable properties, such as query relevance (Xu & Lapata, 2022) and faithfulness (Durmus et al., 2020). We would also like to parametrize the oracle distribution and estimate it from data, so as to derive even more accurate sentence labels.

ACKNOWLEDGEMENTS

We thank the anonymous reviewers for their feedback and Shucong Zhang for useful discussions about the problem formulation. We gratefully acknowledge the support of the UK Engineering and Physical Sciences Research Council (grant EP/W002876/1).

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

## A    GREEDY SEARCH ALGORITHM

---

**Algorithm 2** Labeling with Greedy Search

---
1: **function** GREEDY($n$)          ▷ Max number of sentences in summary
2:       Initialize hypothesis $b$ to empty
3:       **for** $j \leftarrow n$ **do**
4:             Initialize hypothesis score $v$ to 0
5:             **for** $i \leftarrow |D|$ **do**
6:                   $b' = b + \{i\}$, $v' = $ ROUGE($b'$)
7:                   **if** $v' > v$ **then**
8:                         $b \leftarrow b'$, $v \leftarrow v'$
9:             **if** $v = 0$ **then**                              ▷ Early stopping
10:                 **break**
11:       Initialize $\ell_i$ to 0, $\forall i$                    ▷ Multi-hot labels
12:       **for** $x_i \leftarrow b$ **do**
13:             $\ell_i \leftarrow 1$
14:       **return** $\ell$
15: **end function**

---

## B    LABEL STATISTICS

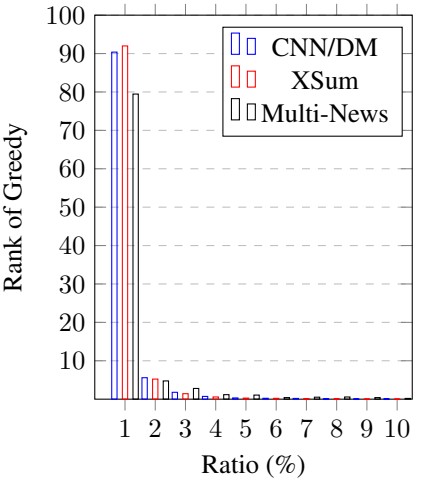

Figure 3: Distribution of greedy oracles over top beams across three validation sets.

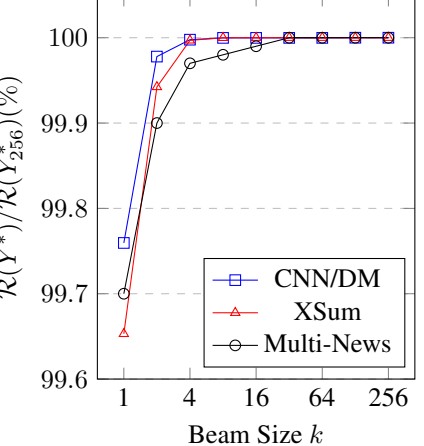

Figure 4: Relative quality of best beams for beam labeling; results shown for different beam sizes across three validation sets.

**Quality of Greedy Labels** Figure 3 shows the distribution of greedy oracles over position in beam search results, as ranked by $\mathcal{R}$ ($k = 256$). As we can see, around 8–20% of greedy labels are not top-ranked, and could thus improve with beam labels. However, as shown in our experimental results, this improvement does not lead to a summarization system that is consistently better across different tasks.

**Effects of Beam Size** As beam search does not guarantee a global optimum either, we further calculate $\mathcal{R}(Y^*)/\mathcal{R}(Y_{256}^*)$ to evaluate the *relative* quality of the top beam $Y^*$ (the top beam found by varied beam sizes), compared against $Y_{256}^*$ (the top beam found by beam size 256). Figure 4 shows that the quality of the top beam converges when beam size increases to 64. However, as oracles are not necessarily indicative of actual model performance (see Section 5.6 for details), we view beam size as a hyperparameter for optimization.

## C EQUIVALENCE PROOF

Given input document $D$, sentence-level inference of a non-autoregressive summarization model $\theta$ is conditionally independent, and the likelihood of an oracle summary $Y^* = \{y_i^*\}_{i=1}^m$ (in its multi-hot representation over $m$ document sentences) is calculated as:

$$p_\theta(Y^*|D) = \prod_{i=1}^m p_\theta(x_i = y_i^*|D). \tag{7}$$

In this case, we show that maximizing the oracle expectation for all sentences is equivalent to the objective in Equation (2):

$$\max \prod_{i=1}^m p_\theta(x_i = \ell_i'|D, S) = \prod_{i=1}^m \mathop{\mathbb{E}}_{Y^* \sim p(Y^*|D,S)} [\mathcal{R}(Y^*, S)p_\theta(x_i|Y^*, D)] \qquad \triangleright \text{ Plug in OREO}$$

$$= \mathop{\mathbb{E}}_{Y^* \sim p(Y^*|D,S)} \left[\mathcal{R}(Y^*, S) \prod_{i=1}^m p_\theta(x_i = y_i^*|D)\right] \qquad \triangleright \text{ Take out } \mathbb{E}$$

$$= \mathop{\mathbb{E}}_{Y^* \sim p(Y^*|D,S)} [\mathcal{R}(Y^*, S)p_\theta(Y^*|D)] . \qquad \triangleright \text{ Apply Eq. (7)}$$

$$\tag{8}$$

We note that our objective in Equation (2) serves as a lower bound of the classic extractive summarization objective $\max p_\theta(Y^*|D)$ *weighted* by oracle evaluation:

$$\max \mathop{\mathbb{E}}_{Y^* \sim p(Y^*|D,S)} [\mathcal{R}(Y^*, S)p_\theta(Y^*|D)] \tag{9}$$

$$= \sum_{Y^*}^{\mathbb{Y}} p(Y^*|D, S)\mathcal{R}(Y^*, S)p_\theta(Y^*|D) \tag{10}$$

$$\leq \mathcal{R}(Y_{\text{best}}^*, S)p_\theta(Y_{\text{best}}^*|D) \text{ where } Y_{\text{best}}^* = \mathop{\arg\max}_{Y^* \in \mathbb{Y}} \mathcal{R}(Y^*, S) \tag{11}$$

$$\propto p_\theta(Y_{\text{best}}^*|D) \tag{12}$$

The equality holds only if the oracle distribution $p(Y^*|D, S)$ is a Dirac delta distribution $\delta(Y^* - Y_{\text{best}}^*)$.

## D EFFECTS OF ORACLE DISTRIBUTION

Table 9: OREO results with different oracle distributions on the CNN/DM validation set.

| Systems | ROUGE-1 | ROUGE-2 | ROUGE-L |
|---|---|---|---|
| BERTSUM | | | |
|     Greedy | 44.00 | 20.73 | 40.45 |
|     OREO, $U(1, t)$ | **44.26** | **20.92** | **40.69** |
| OREO, $A_r(1, 16)$ | 44.06 | 20.76 | 40.50 |
| OREO, $A_q(1, 16)$ | 44.17 | 20.83 | 40.60 |
| OREO, $A_\ell(1, 16)$ | 44.03 | 20.73 | 40.42 |
| OREO, $A_p(1, 16)$ | 43.91 | 20.45 | 40.26 |

We devise and experiment with several oracle distributions that assign non-uniform probability to top $t$ beams:

1. **Annealing over Rank** $A_r$ decreases the unnormalized weight from 1 to 0 over top beams, assuming that the oracle distribution positively correlates with *hypothesis rank*.

2. **Annealing over Quality** $A_q$ sets the unnormalized weight for a top beam $Y^*$ as $\mathcal{R}(Y^*)$, assuming that the oracle distribution positively correlates with *hypothesis evaluation score*.

3. **Annealing over Locality** $A_\ell$ defines the locality of a hypothesis $Y$ to be proportional to the mean of sentence-level scores $\overline{\mathcal{R}(y_j^*, S)}$, $y_j^* \in Y^*$. This is based on the assumption that a hypothesis is more *local* if its sentences are, by themselves, high-scoring. Hypothetically, these sentences stand a higher chance to be selected by a non-autoregressive sequence labeling model which presumably focuses more on their individual features rather than collective information (Zhong et al., 2020).

4. **Annealing over Position Rank** $A_p$ decreases the unnormalized weight from 1 to 0 over top beams which are reversely ranked by their position in the original document, assuming that oracle distribution positively correlates with *document position*.

Table 9 presents our results on CNN/DM validation set. As we can see, the above-mentioned distributions do not yield better results than simply adopting a uniform distribution (third row). We believe this is because hand-crafted distributions are all associated with strong assumptions, which may not be valid for real-world summarization data. However, we note that most of these distributions still manage to outperform greedy labels, showing consistent gains when considering information from multiple oracles.

Apart from heuristic oracle distributions, we could also learn a parametrized distribution from data. For instance, a model with a uniform oracle distribution could be trained to derive a potentially more accurate estimation from its predictions. A new set of sentence labels would be then calculated with Equation (5), and used to improve the optimization of a new model.

## E    DETAILS OF EXPERIMENTAL SETTINGS

Table 10: Hyperparameters for supervised training of BERTSUM on five summarization datasets.

| Monolingual | CNN/DM | XSum | Multi-News | Reddit | WikiHow |
|---|---|---|---|---|---|
| Beam size $k$ | 256 | 16 | 16 | 256 | 16 |
| Oracle distribution $t$ | 16 | 16 | 16 | 32 | 16 |

**Monolingual Extractive Summarization**    We used three GeForce RTX 2080 GPUs for model training and `bert.base` in our experiments. We refer interested readers to Liu & Lapata (2019) for detailed training configurations which are identical to ours. Following Liu & Lapata (2019), we used the Python package `pyrouge` for calculating ROUGE. For our the proposed labeling methods, we searched over the following $(k, t)$ pairs: $(256, 32), (256, 16), (256, 8), (32, 32), (16, 16), (8, 8)$. We show the best-performing hyperparameter combinations for each dataset in Table 10. We used standard parameter settings for all experiments: ROUGE-1.5.5.pl -c 95 -m -r 1000 -n 2 -a. We used the datasets as preprocessed by Zhong et al. (2020) which can be accessed at: `https://github.com/maszhongming/matchsum`.

**Cross-Lingual Extractive Summarization**    In our cross-lingual experiments we used four GeForce RTX 2080 GPUs for model training with `xlmr.base` and `mbert.base`. Particularly, interval embeddings (Liu & Lapata, 2019) were used in MBERTSUM but not in XLS since XLM-R removes segment embeddings from the input following RoBERTA (Liu et al., 2019). We refer readers to Jia et al. (2022) for details on training configuration; we made minimal adjustments to adapt to our training environment, i.e., no training hyperparameters were specifically optimized for our method. We set the batch size to 4, and accumulated gradients every 32 steps. Following Jia et al. (2022), we used word replacement rate of 0.5 to learn cross-lingual representation alignment. We fine-tuned models on the English data with a learning rate of $2 \times 10^{-3}$ for 50,000 optimization steps, and a warm-step of 10,000. Following Jia et al. (2022), we used the Python package `spacy` for non-English hypothesis/reference tokenization, and `pyrouge` for ROUGE calculation.

**Abstractive Summarization**    In our abstractive summarization experiments we used four GeForce RTX 2080 GPUs for model training with `bart.large`; the latter was also used in our baseline BART system and to initialize GSUM. Due to GPU memory limitations, we set the maximum length of an input document to 640 tokens (with the excess clipped) and used half float precision for efficient

training. We used one sample for each GPU, and accumulated gradients every 32 steps. We fine-tuned all models on CNN/Daily Mail with a learning rate of $3 \times 10^{-5}$ for 20,000 optimization steps, and a warm-step of 500. Following the evaluation steps in BART (Lewis et al., 2020) and GSUM (Dou et al., 2021), we used `file2rouge`[5] to evaluate abstractive summaries.

# F  EXTENDED RESULTS

Table 11: Results for extractive summarization on **CNN/DM** test set. Results for Greedy are taken from Zhong et al. (2020).

| Systems | R-1 | R-2 | R-L |
|---|---|---|---|
| MATCHSUM (Bert-based) | | | |
| Greedy | 44.22 | 20.62 | 40.38 |
| OREO | **44.32** | **20.66** | **40.51** |
| MATCHSUM (Roberta-based) | | | |
| Greedy | 44.41 | **20.86** | 40.55 |
| OREO | **44.49** | 20.84 | **40.63** |

Table 12: Results of BERTSUM with *local labeling* on **CNN/DM** test set.

| Systems | R-1 | R-2 | R-L |
|---|---|---|---|
| Local (soft) | 42.45 | 19.55 | 38.75 |
| Local (top3) | 42.59 | 19.67 | 38.87 |
| Local (top5) | 42.39 | 19.47 | 38.67 |
| OREO | **43.58** | **20.43** | **39.96** |

**Supervised Extractive Summarization**    We next investigate whether OREO can further improve performance when integrated with MATCHSUM (Zhong et al., 2020). To provide a proof of concept, we further built MATCHSUM + OREO, using BERTSUM (OREO) as a sentence ranker at inference. We experimented with two MATCHSUM versions (based on either BERT or RoBERTA), and show the results on CNN/DM in Table 11. Note that MATCHSUM+OREO uses the off-the-shelf summary-level reranker from MATCHSUM, i.e., it is not retrained on predictions from BERTSUM trained on OREO. As we can see, OREO improves upon vanilla MATCHSUM simply on account of introducing higher-quality candidate sentences, and thus higher-quality candidate summaries for reranking. Retraining the summary-level reranker to avoid input distribution shift would result in additional performance gains.

We further experimented with *local labeling* (Cheng & Lapata, 2016) and the results on CNN/DM are shown in Table 12. Local (soft) labels sentences with max-min scaled ROUGE scores (which are soft). In contrast, top3 and top5 create discrete scores via assigning positive labels to the top3 and top5 sentences, and all other sentences are given negative labels. As we can see, local labeling schemes are inferior to OREO, underscoring the need to exploit summary-level information for sentence annotation.

Table 13: Cross-lingual zero-shot performance on test sets of **MLSum** in ROUGE-L.

| Systems | De | Es | Fr | Ru | Tr | AVG |
|---|---|---|---|---|---|---|
| MBERTSUM | | | | | | |
| Greedy | *22.68* | *20.44* | *22.70* | *8.71* | *27.89* | *20.48* |
| Beam | 28.36 | 20.55 | 22.74 | **9.30** | 29.38 | 22.07 |
| OREO | **29.13** | **20.62** | **22.82** | 9.13 | **30.78** | **22.50** |

**Cross-Lingual Extractive Summarization**    We further initialize BERTSUM with mBERT (Devlin et al., 2019). As we can in Table 13, MBERTSUM finetuned with greedy labels shows inferior performance across languages. Nevertheless, OREO leads to substantial performance gains on both German and Turkish (we observe a similar trend when BERTSUM initialized with XLM-R).

---

[5]https://github.com/pltrdy/files2rouge

Table 14: Results for abstractive summarization on **CNN/DM** test set.

| Systems | ROUGE-1 | ROUGE-2 | ROUGE-L |
|---|---|---|---|
| GSUM trained with greedy ORACLE guidance | | | |
| Greedy | *44.40* | *21.52* | *41.23* |
| Beam | 44.41 | 21.55 | 41.26 |
| OREO | **44.48** | **21.60** | **41.33** |
| GSUM trained with OREO ORACLE guidance | | | |
| Greedy | 44.66 | 21.72 | 41.43 |
| Beam | 44.68 | 21.74 | 41.47 |
| OREO | **44.81** | **21.83** | **41.60** |

**Supervised Abstractive Summarization** We show extended abstractive results in Table 14. The first block shows performance of vanilla GSUM (Dou et al., 2021) which uses greedy extractive oracles as guidance during training. During inference, we compare to three types of extractive guidance produced by BERTSUM trained with greedy, beam, and OREO labels. Despite the training-testing discrepancy in guidance, extractive guidance with OREO labels helps generate better downstream abstractive summaries.

To further validate the effectiveness of OREO on the optimization of GSUM, we trained GSUM with OREO ORACLE, and the results are shown in the second block. As we can see, adopting OREO guidance during training further boosts system performance, while the system using OREO for both training and testing achieves the best results, i.e., 0.37 ROUGE-L improvement over GSUM.

Table 15: Performance on four recently proposed metrics for summary evaluation (**CNN/DM** test set). Results shown for extractive (BERTSUM) and abstractive (GSUM) summarization.

| Systems | ROUGE-WE | MOVERSCORE | BERTSCORE | SUMMAQA |
|---|---|---|---|---|
| BERTSUM | | | | |
| Greedy | 33.04 | 27.66 | 87.72 | 27.13 |
| Beam | 33.09 | 27.72 | 87.71 | 26.91 |
| OREO | **33.41** | **27.94** | **87.77** | **27.15** |
| GSUM | | | | |
| Greedy | 34.72 | 29.17 | 88.83 | 32.11 |
| Beam | 34.75 | 29.20 | 88.83 | 32.11 |
| OREO | **35.06** | **29.49** | **88.89** | **32.20** |

**Performance on Other Metrics** Despite being popular in summarization, ROUGE has its weaknesses in summary evaluation. Therefore, we further evaluated the generated summaries on CNN/DM using other recently proposed metrics: ROUGE-WE (Ng & Abrecht, 2015), MOVERSCORE (Zhao et al., 2019), BERTSCORE (Zhang et al., 2020b), and SUMMAQA (Scialom et al., 2019). We show the results in Table 15.

ROUGE-WE extends ROUGE using soft lexical matching based on the cosine similarity of Word2Vec. We report the average of N-gram results ($N = \{1, 2, 3\}$) for ROUGE-WE. BERTSCORE computes similarity scores between reference tokens and generated tokens but with BERT embeddings. MOVER-SCORE uses Word Mover's Distance to measure the semantic distance between a reference and a hypothesis summary. SUMMAQA is a QA-style metric which was initially proposed to generate questions from source documents. Following the evaluation of Papalampidi & Lapata (2022), we apply the question generation model to reference summaries to obtain more relevant and informative questions. As we can see, OREO achieves better performance across metrics for both extractive and abstractive summarization.

## G   SYSTEM OUTPUT

**Document**: The largest single high-definition map of mysterious dark matter has been produced. It is the first in a series of maps of the cosmos that will eventually allow a 3D view of dark matter across one eighth of the night sky. And the map should allow astronomers to study how galaxies formed in the universe. University of Manchester researchers have revealed an HD dark matter map (shown). It shows clumps of mystery particles across 0.4 per cent of the sky. The goal is to eventually map 12.5 per cent over five years. Red here shows more dark matter, and blue shows less. The moon is shown top left for scale. A team from the University of Manchester, led by Dr Sarah Bridle, has spent the past two years measuring the shapes of galaxies used to construct the map. And the map was released today at the April meeting of the American Physical Society in Baltimore, Maryland. It was created using data captured by the Dark Energy Camera, a 570-megapixel imaging device that is the primary instrument for the Dark Energy Survey (DES) led by Fermilab in the US...

**Reference Summary**: University of Manchester researchers reveal HD dark matter map. It shows clumps of mystery particles across 0.4 per cent of the sky. The goal is to eventually map 12.5 per cent over five years. Results could help reveal how galaxies form in the universe.

---

Extractive Summarization with BERTSUM

**Greedy**: University of Manchester researchers have revealed an HD dark matter map (shown). It shows clumps of mystery particles across 0.4 per cent of the sky. It is the first in a series of maps of the cosmos that will eventually allow a 3d view of dark matter across one eighth of the night sky.

**Beam**: University of Manchester researchers have revealed an HD dark matter map (shown). The goal is to eventually map 12.5 per cent over five years. It is the first in a series of maps of the cosmos that will eventually allow a 3d view of dark matter across one eighth of the night sky.

**OREO**: University of Manchester researchers have revealed an HD dark matter map (shown). It shows clumps of mystery particles across 0.4 per cent of the sky. The goal is to eventually map 12.5 per cent over five years.

---

Abstractive Summarization with GSUM

**Greedy**: University of Manchester researchers have revealed an HD dark matter map. It shows clumps of mystery particles across 0.4 per cent of the sky. The goal is to eventually map 12.5 per cent over five years. It is the first in a series of maps of the cosmos that will eventually allow a 3D view of dark matter across one eighth of the night sky. And the map should allow astronomers to study how galaxies formed in the universe.

**Beam**: University of Manchester researchers have revealed an HD dark matter map. It shows clumps of mystery particles across 0.4 per cent of the sky. The goal is to eventually map 12.5 per cent over five years. It is the first in a series of maps of the cosmos that will eventually allow a 3D view of dark matter across one eighth of the night sky. And the map should allow astronomers to study how galaxies formed in the universe.

**OREO**: University of Manchester researchers have revealed an HD dark matter map of the cosmos. It shows clumps of mystery particles across 0.4 per cent of the sky. Red here shows more dark matter, and blue shows less. The goal is to eventually map 12.5 per cent over five years. And the map should allow astronomers to study how galaxies formed in the universe.

---

Table 16: Examples of system output on the CNN/DM test set. We illustrate differences among labeling algorithms with a sentence from the reference summary labeled in red. BERTSUM trained with OREO labels includes the sentence in its extract. In contrast, Greedy selects a suboptimal, verbose sentence highlighted in blue, potentially due to its position in the beginning of the original document (lead bias). The Beam extract includes both sentences and is therefore most redundant. Using these extracts as inference guidance, GSUM creates abstractive summaries which for Greedy and Beam are identically verbose, while OREO summary is more concise.

# H    BOUND PRESERVATION

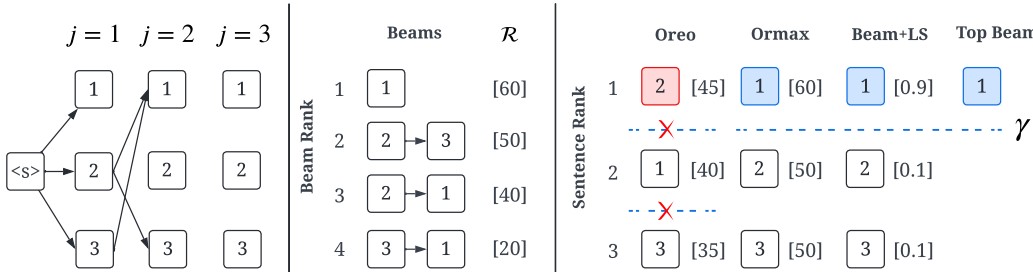

Figure 5: Illustration of beam search (left; early stopped at $j = 2$), ranked beams (middle), and ranked sentences (right). We show sentences (in squares) and their scores (in brackets) under different labeling algorithms. For OREO, there does not exist a $\gamma$ that halves the ranked list in a way that the top half is identical to the sentences selected by the top beam.

