# OpenReview forum: "Text Summarization with Oracle Expectation"
_ICLR.cc/2023/Conference — ICLR 2023 poster_

### Official Review · Reviewer_mPYc · 2022-10-24

**Confidence:** 3
**Correctness:** 4
**Technical Novelty And Significance:** 3
**Empirical Novelty And Significance:** 2
**Recommendation:** 6

**Clarity, Quality, Novelty And Reproducibility:**

Clarity: the paper is clear and well written.

Novelty: the work is novel to the best of my knowledge.

Quality: both theoretical and empirical aspects of the paper are good.

Reproducibility: the code and models are open-sourced (I could not check given broken anonymized link).



**Details Of Ethics Concerns:**

If this paper is from Mirella Lapata's group (as I suspect given its citations), there is a potential conflict of interest for me reviewing given the Professor's industry collaborations (although I have never worked with them and do not know them).

**Strength And Weaknesses:**

Overall, this paper is well written and well structured. Although the contribution is fairly simple, it is well theorized. Extensive experiments are also provided.

This work is likely to have little practical use given the additional complexity of the proposed scheme and its fairly limited experimental improvements. But it does provide interesting insights and may trigger further research.

Suggestions/Comments:
- can improvements that are statistically significant be added to the result tables? Many improvements seem limited; are they statistically significant?
- since the method does not yield state-of-the-art results, it would be nice to better explain summary reranking and MatchSum (e.g., how much more complicated/computationally expensive these are).
- could MatchSum be added as a baseline for the cross-domain experiments?

**Summary Of The Paper:**

This paper introduces a novel labeling scheme for extractive text summarization. Extractive summarization datasets are most often derived from abstractive datasets using a greedy labeling scheme, providing a single extractive reference per document. Greedy multi-hot labels cause sparsity/under-fitting issues so the authors propose a soft-labeling scheme and show it is equivalent to incorporating multiple hypotheses in a non-autoregressive labeling model. The soft-labeling oracle is implemented using beam search. The authors then show improvements over greedy and beam labeling in several settings, although the model itself is not state-of-the-art.

**Summary Of The Review:**

The work is novel to the best of my knowledge but its significance is likely to be limited given:
- recent improvements to abstractive summarization
- its complexity over labeling schemes that are simpler to implement
- it is not state-of-the-art

That said, the work is overall solid and well theorized.

---

> ### Author Response · Authors · 2022-11-17
> **Response to Reviewer mPYc**
>
> > Q1: This work is likely to have little practical use given the additional complexity of the proposed scheme and its fairly limited experimental improvements. But it does provide interesting insights and may trigger further research.
>
> Response: we agree that the past few years have witnessed a surge of interest in abstractive summarization partially due the availability of large pre-trained models. Nevertheless,  extractive summarization remains a practical approach especially in industrial settings where hallucinations should be minimized. We motivate our method from an optimization perspective, but the actual implementation is quite simple. We have included the code in our submission. Due to the lack of standard sentence labeling tools for summarization, we also plan to release an open-source python package for this. It will support different existing labeling schemes (greedy, beam, Oreo etc) and help users  obtain extractive labels painlessly.
>
> > Q2: Can improvements that are statistically significant be added to the result tables? Many improvements seem limited; are they statistically significant?
>
> Response: In the following table, we show the datasets on which Oreo performs significantly better than greedy/beam labeling (with 95% confidence interval via bootstrap resampling [1]):
>
> |Tasks                   |    Greedy    |     Beam    |
> | ---------------------- | ------------ | ----------- |
> |Supervised Extractive   | CD, XS, RD   | CD, XS, WH  |
> |Zero-Shot Cross-Domain  | RD           | WH          |
> |Zero-Shot Cross-Lingual | De, Tr, AVG  | De, Tr, AVG |
> |Supervised Abstractive  | CD           | CD          |
>
> For example, on supervised extractive summarization, Oreo’s ROUGE scores are significantly different from greedy labeling on CNN/DM, XSum, and Reddit; and from Beam search on CNN/DM, XSum, and WikiHow. As we can see, Oreo achieves significant improvements on many tasks and datasets.
>
> > Q3: Since the method does not yield state-of-the-art results, it would be nice to better explain summary reranking and MatchSum (e.g., how much more complicated/computationally expensive these are).
>
> Response: thank you for the comment, and we will explain MatchSum in detail in the paper. We have also achieved better results than MatchSum; please see our response to Reviewer Sdw2 (Q3).
>
> > Q4: Could MatchSum be added as a baseline for the cross-domain experiments?
>
> Response: We further evaluated MatchSum in cross-domain settings and the results are as follows (where best results are highlighted with **):
>
> | Models           | XS        | RD        | WH        |
> | ---------------- | --------- | --------- | --------- |
> | MatchSum         | **15.75** | **17.82** | 25.10     |
> | BertSum (Greedy) | 15.62     | 17.06     | 25.39     |
> | BertSum (Oreo)   | 15.58     | 17.71     | **25.62** |
>
> As we can see, MatchSum improves over BertSum (with Greedy and Oreo labels) on XSum and Reddit, but not on WikiHow where BertSum (Oreo) performs best. We will include these results in the revised version of the paper.
>
> ### References
>
> 1. *Bootstrap methods and their application* (No. 1). Cambridge university press.

---

### Official Review · Reviewer_perC · 2022-10-24

**Confidence:** 4
**Correctness:** 3
**Technical Novelty And Significance:** 2
**Empirical Novelty And Significance:** 3
**Recommendation:** 6

**Clarity, Quality, Novelty And Reproducibility:**

Clarity - The paper is mostly clear, except that some equations are a bit hard to understand
Quality - Quality needs improvement in terms of more experiments with different models
Novelty - Somewhat novel
Reproducibility - The work is reproducible and the codes are available.


**Strength And Weaknesses:**

Strengths
The problem statement is very interesting.
Proposed method is simple to implement and doesn’t require architectural changes for existing models
Extensive set of experiments on different settings and on multiple datasets performed, showing the method leads to better-performing models
Weaknesses and suggestions
Only the BERTSum model was used for comparison between labelling strategies. Performing similar experiments with some other models would be better, and these models need not be very complex (even some older architectures such as SummaRuNNer [Nallapati et al., 2017] may be used).
Another simple labelling strategy that may be a baseline may be to just get the top-K ROUGE sentences from the document individually for every sentence in the original reference summary. [Cheng & Lapata (2016)]
In Zero shot multi-lingual setting XLM-base has been used when the given SOTA uses XLM-Large, but then the authors claim the SOTA performs better because of the more complex architecture.
Some equations are a bit difficult to understand, and it will be good if the terms are defined properly and described in simpler text.
The purpose of Section 5.5 is not very clear, and the results do not seem too interesting either. It is also not clear how exactly the ASK metric measures the extent to which a model can attain summary relevant knowledge. This space may be better used for discussing experiments using other models.
In many of the results, only a very small improvement in ROUGE scores have been achieved over other labelling strategies (which is kind of expected with ROUGE scores). It is also known that ROUGE scores are probably not a very good way of measuring quality of summaries. Thus it is not very evident if the OREO strategy actually leads to better summaries. It would perhaps be better if some survey analyses were performed with the outputs from models trained on different labelling strategies.

Other Comments:
There is some discrepancy with the caption and data in Table 1, it seems the authors talk about sentences 1 & 4 (not 2 & 4) but then the OREO does not provide very high score (relatively) to sentence 1.
In line 8 of Algorithm 1, the l’ on the right hand side will probably be x.


**Summary Of The Paper:**

The paper proposes a novel way of extracting silver-standard extractive summaries (Oracle) from gold-standard abstractive summaries which they call “OREO”. Earlier methods mostly relied on the Greedy labelling method, which is sub-optimal and deterministic. The authors propose to instead generate the final oracle summaries by soft labelling so that multiple possible extractive oracles (obtained by beam-search) are considered according to their probability distribution; which leads to the final oracle summary being non-deterministic.
The authors have then performed several experiments to show the robustness of their model in different settings - supervised, zero-shot cross-domain, zero-shot cross-lingual, supervised abstractive, etc. The authors have experimented on several datasets, using the BERTSum model (and GSum model for abstractive setting). They show that models trained on OREO perform better than greedy and simple-beam search labels in most cases and generalize better.

**Summary Of The Review:**

While the proposed method is simple to implement and is shown to be robust by several experiments, my main problem is that only BERTSum model was used to evaluate the different labelling strategies. Sections 5.5 and 5.6 are also questionable. Even though the method performs better than others, in most cases the increase in performance is quite small.

---

> ### Author Response · Authors · 2022-11-17
> **Response to Reviewer perC (1/2)**
>
> > Q1: Only the BERTSum model was used for comparison between labelling strategies. Performing similar experiments with some other models would be better, and these models need not be very complex (even some older architectures such as SummaRuNNer [Nallapati et al., 2017] may be used).
>
> Response: thank you for the suggestion. We further conducted experiments on SummaRuNNer and show the results below (we highlight best results with **):
>
> | Labeling                        | R-1       | R-2       | R-L       |
> | ------------------------------- | --------- | --------- | --------- |
> | Greedy (Nallapati et al., 2017) | 39.60     | 16.20     | 35.30     |
> | Greedy (our implementation)     | 41.55     | 19.05     | 38.00     |
> | Oreo (our implementation)       | **42.01** | **19.12** | **38.31** |
>
> Note that the original implementation of SummaRuNNer is not publicly available. In our implementation, we observe a considerable performance improvement, possibly due to different data preprocessing steps (we used the processed data from Zhong et al. (2020); see Appendix E for details). In our implementation, SummaRuNNer with Oreo yields better performance on CNN/DM.
>
> > Q2: Another simple labelling strategy that may be a baseline may be to just get the top-K ROUGE sentences from the document individually for every sentence in the original reference summary. [Cheng & Lapata (2016)]
>
> Response: We further experimented with the suggested labeling scheme (which we call *local labeling* in the paper) and the results on CNN/DM are  as follows (we highlight best results with **):
>
> | Labeling     | R-1       | R-2       | R-L       |
> | ------------ | --------- | --------- | --------- |
> | Local (soft) | 42.45     | 19.55     | 38.75     |
> | Local (top3) | 42.59     | 19.67     | 38.87     |
> | Local (top5) | 42.39     | 19.47     | 38.67     |
> | Oreo         | **43.58** | **20.43** | **39.96** |
>
> Local (soft) labels sentences with max-min scaled ROUGE scores (which are soft). In contrast, top3 and top5 create discrete scores via assigning positive labels to the top3 and top5 sentences, and all other sentences are given negative labels. As we can see, local labeling schemes are inferior to Oreo, underscoring the need to exploit summary-level information for sentence annotation.
>
> > Q3: In Zero shot multi-lingual setting XLM-base has been used when the given SOTA uses XLM-Large, but then the authors claim the SOTA performs better because of the more complex architecture.
>
> Response: SOTA (NLS) uses 1) XLMR-large, 2) a few customized layers which introduce more parameters for neural label searching (NLS), and 3) a machine translation system for sentence labeling. We did not use any of these but we believe the gap with SOTA can be further reduced if we do.
>
> > Q4: Some equations are a bit difficult to understand, and it will be good if the terms are defined properly and described in simpler text.
>
> Response: thank you for the comment. We will further elaborate on the equations in the revised version of this paper.

---

> ### Author Response · Authors · 2022-11-17
> **Response to Reviewer perC (2/2)**
>
> > Q5: The purpose of Section 5.5 is not very clear, and the results do not seem too interesting either. It is also not clear how exactly the ASK metric measures the extent to which a model can attain summary relevant knowledge.
>
> Response: we would like to provide some motivation for Section 5.5 and why we think it can be helpful to future work in this line of research.
>
> - Oreo is, fundamentally, a soft labeling scheme for extractive summarization. Although soft labeling is understudied in extractive summarization, label smoothing (LS), a simple strategy to create soft labels, has been proved effective in many other learning tasks.
> - After outperforming hard labels in various experimental settings with Oreo, Section 5.5 provides a comparison amongst soft labeling methods. Such a comparison is desirable as our community may not need Oreo if a simple soft labeling method such as LS can achieve superior performance.
> - However, we empirically show that LS does not work very well on extractive summarization. A fundamental difference between LS and Oreo is bound preservation, which intuitively should be a nice property to have. Nevertheless, based on a comparison between LS, Oreo, and its bound-preserving variant Ormax, we show that the upper bound fails to faithfully indicate actual model performance.
> - In this context, Section 5.6 aims to clarify why this is the case (i.e., why soft labels with high upper bound deliver inferior results). We suspect the reason is the constrained inference setting for extractive summarization: during inference, a *fixed* number of sentences have to be extracted *independently*. This may render the upper bound extremely relaxed and therefore not indicative. To quantify this gap, we use a sampling-based approach to imitate this inference setting, leading to what we call attainable summary knowledge (ASK). Compared to the upper bound, we show that ASK is more consistent with our empirical results, and we hope this insight can motivate the development of better labeling strategies, as endorsed by Reviewer Sdw2.
>
> > Q6: It is also known that ROUGE scores are probably not a very good way of measuring quality of summaries. Thus it is not very evident if the OREO strategy actually leads to better summaries. It would perhaps be better if some survey analyses were performed with the outputs from models trained on different labelling strategies.
>
> Response: Thank you for the suggestion; we agree that ROUGE, despite being popular in summarization, has its weaknesses in summary evaluation. Therefore, we further evaluated the generated summaries on CNN/DM using other recently proposed metrics: ROUGE-we [1], MoverScore [2], BERTScore [3], and SummaQA [4] (we highlight best results with **).
>
> BertSum (Extractive) Results:
>
> | Labels | ROUGE-we  | MoverScore | BERTScore | SummaQA  |
> | ---------- | --------- | ---------- | --------- | --------- |
> | Greedy| 33.04     | 27.66      | 87.72     | 27.13     |
> | Best Beam  | 33.09     | 27.72      | 87.71     | 26.91     |
> | Oreo | **33.41** | **27.94**  | **87.77** | **27.15** |
>
> GSum (Abstractive) Results:
>
> | Labels | ROUGE-we  | MoverScore | BERTScore | SummaQA  |
> | ----------- | --------- | ---------- | --------- | --------- |
> | Greedy | 34.72 | 29.17 | 88.83     | 32.11     |
> | Best Beam   | 34.75     | 29.20      | 88.83     | 32.11     |
> | Oreo | **35.06** | **29.49** | **88.89** | **32.20** |
>
> ROUGE-we extends ROUGE using soft lexical matching based on the cosine similarity of Word2Vec. We report the average of N-gram results (N={1,2,3}) for ROUGE-we. BertScore computes similarity scores between reference tokens and generated tokens but with BERT embeddings. MoverScore uses Word Mover’s Distance to measure the semantic distance between a reference and a hypothesis summary. SummQA is a QA-style metric which was initially proposed to generate questions from source documents. Following the evaluation of [5], we apply the question generation model to reference summaries to obtain more relevant and informative questions. As we can see, Oreo achieves better performance across metrics for both extractive and abstractive summarization.
>
> > Q7: There is some discrepancy with the caption and data in Table 1, it seems the authors talk about sentences 1 & 4 (not 2 & 4) but then the OREO does not provide very high score (relatively) to sentence 1. In line 8 of Algorithm 1, the l’ on the right hand side will probably be x.
>
> Response: thank you for spotting these typos! We will fix them in the revised version.
>
> ### References
>
> 1. *Better Summarization Evaluation with Word Embeddings for ROUGE*, EMNLP’15
> 2. *BERTScore: Evaluating Text Generation with BERT*, ICLR’20
> 3. *MoverScore: Text Generation Evaluating with Contextualized Embeddings and Earth Mover Distance*, EMNLP’19
> 4. *Answers Unite! Unsupervised Metrics for Reinforced Summarization Models*, EMNLP’19
> 5. *Hierarchical3D Adapters for Long Video-to-text Summarization*, arXiv preprint arXiv:2210.04829

---

### Official Review · Reviewer_Sdw2 · 2022-10-26

**Confidence:** 3
**Correctness:** 4
**Technical Novelty And Significance:** 4
**Empirical Novelty And Significance:** 3
**Recommendation:** 8

**Clarity, Quality, Novelty And Reproducibility:**

The paper is well organized and presented. Though soft labels are widely used in modern deep-learning models, incorporating them in a sentence labeling process is a novel design. The authors also provide a theoretical explanation of the learning objective involving soft labels. In addition to experimenting with the labeling method in various text summarization scenarios, the analyses on the connection between the upper bound of different oracles and the actual performance (as well as the proposed attainable summary knowledge) are inspirational. The authors release their code for reproducibility.

A minor concern is that performance improvements over baselines or deterministic beam/greedy oracle seem marginal in some cases. It's weird to mark the results of OREO in Table 4 in bold since MatchSum is the most performance model. The authors claim that MatchSum has an extra reranking process, and OREO can potentially serve as a foundation for those more complex methods. I would like to know whether OREO can further enhance performance over MatchSum. Meanwhile,  the idea of using reinforcement learning over evaluation metrics or exploiting supervision signals from multiple target summaries can be widely found in abstract summarization efforts (e.g., [1][2]), and some of them may not be hard to adapt to extractive summarization. Comparing these works can make the evaluation more solid.

I'm also curious why the evaluation metric considers ROUGE-1 and ROUGE-2 but not ROUGE-L. Will the experimental conclusions still hold if using the mean of ROUGE-1, 2, and L?

[1] Deep Reinforcement Learning with Distributional Semantic Rewards for Abstractive Summarization. EMNLP 2019

[2] BRIO: Bringing Order to Abstractive Summarization. ACL 2022


**Strength And Weaknesses:**

**Strengths**
- The identified problem is significant in extractive summarization, and the motivation is clear.
- The idea of using soft labels for sentence-level labels is novel.
- Extensive experiments and thorough analyses demonstrate the superiority of the proposed method in many summarization scenarios across domains and languages in supervised and zero-shot settings, with
- A theoretical proof is provided, showing that maximizing oracle expectation is equivalent to the proposed non-deterministic objective for extractive summarization.

**Weaknesses**
- The performance improvements look marginal in some settings, and more sophisticated baselines can be considered.


**Summary Of The Paper:**

This paper addresses two common problems in pseudo-label (or oracle extract) generation for extractive summarization: 1) greedy or beam search does not always lead to a globally optimal solution, and 2) a pseudo-label is a deterministic training target that lacks calibration.
The authors then propose to leverage a series of pseudo-summaries generated from beam search to generate sentence-wise soft labels used for extractive summarization training.


**Summary Of The Review:**

This paper presents a sentence labeling algorithm creating soft labels for extractive text summarization, with clear motivation, novel technical design, extensive evaluation,  and theoretical analysis.

---

> ### Author Response · Authors · 2022-11-17
> **Response to Reviewer Sdw2**
>
> > Q1: A minor concern is that performance improvements over baselines or deterministic beam/greedy oracle seem marginal in some cases.
>
> Response: we agree that performance gains with Oreo vary for different datasets and tasks, and this is why we conducted extensive experiments across multiple benchmarks and experimental settings, including (cross-lingual) extractive and (monolingual) abstractive summarization. Our results show *consistent* gains over greedy labeling. We further provide a detailed analysis as to what constitutes a good labeling scheme, showing that standard labeling schemes in summarization are not optimal. We hope our work can motivate the development of even better sentence labeling schemes.
>
> > Q2: It's weird to mark the results of OREO in Table 4 in bold since MatchSum is the most performance model.
>
> Response: thank you for pointing this out! We mark these results in bold as they are highest in the BertSum block. We will further clarify this in the table caption.
>
> > Q3: The authors claim that MatchSum has an extra reranking process, and OREO can potentially serve as a foundation for those more complex methods. I would like to know whether OREO can further enhance performance over MatchSum.
>
> Response: thank you for the suggestion. To provide a proof of concept, we further built MatchSum + Oreo,  using BertSum (Oreo) as a sentence ranker at inference. We experimented with two MatchSum versions (based on either Bert or Roberta), and show the results on CNN/DM as follows (we highlight best results with **):
>
> | Systems (Bert-based)    | R-1       | R-2       | R-L       |
> | ----------------------- | --------- | --------- | --------- |
> | MatchSum                | 44.22     | 20.62     | 40.38     |
> | MatchSum + Oreo         | **44.32** | **20.66** | **40.51** |
>
> | Systems (Roberta-based) | R-1       | R-2       | R-L       |
> | ----------------------- | --------- | --------- | --------- |
> | MatchSum                | 44.41     | **20.86** | 40.55     |
> | MatchSum + Oreo         | **44.49** | 20.84     | **40.63** |
>
> Results for MatchSum (Bert) and MatchSum (Roberta) are taken from Zhong et al., 2020. Note that MatchSum+Oreo uses the off-the-shelf summary-level reranker from MatchSum, i.e., it is not retrained on predictions from BertSum trained on Oreo.
>
> As we can see, Oreo improves upon vanilla MatchSum simply on account of introducing higher-quality candidate sentences, and thus higher-quality candidate summaries for reranking. Retraining the summary-level reranker to avoid input distribution shift would result in additional performance gains. This is supported by our experiments in Section 5.4 where the use of Oreo *during training* leads to considerable improvements in abstractive summarization (see Table 12 in Appendix F). Due to time and resource constraints, we leave this to future work.
>
> > Q4: Meanwhile, the idea of using reinforcement learning over evaluation metrics or exploiting supervision signals from multiple target summaries can be widely found in abstract summarization efforts (e.g., [1][2]), and some of them may not be hard to adapt to extractive summarization. Comparing these works can make the evaluation more solid.
>
> Response: we agree these references are related to our work. We will discuss them in the revised version of this paper.
>
> > Q5: I'm also curious why the evaluation metric considers ROUGE-1 and ROUGE-2 but not ROUGE-L. Will the experimental conclusions still hold if using the mean of ROUGE-1, 2, and L?
>
> Response: we follow previous work and use the mean of ROUGE-1 and ROUGE-2 as our evaluation metric. We assume computational efficiency is one of the primary factors behind this choice: ROUGE-L computes the length of the Longest Common Subsequence (LCS) between two strings which, compared to ROUGE-1 and ROUGE-2, has a higher time complexity and is much slower in practice. We show the labeling speed (second/sample; CNN/DM) as follows:
>
> | Labeling | w/o ROUGE-L | with ROUGE-L |
> | -------- | ----------- | ------------ |
> | Greedy   | 0.0024      | 0.0394       |
> | Oreo     | 0.0830      | 0.5259       |
>
> We further evaluated labeling performance with the mean of ROUGE-1, 2, and L:
>
> | Labeling            | R-1   | R-2   | R-L   |
> | ------------------- | ----- | ----- | ----- |
> | Greedy              | 43.17 | 20.13 | 39.56 |
> | Greedy with ROUGE-L | 43.24 | 20.15 | 39.66 |
> | Oreo                | 43.58 | 20.43 | 39.96 |
> | Oreo with ROUGE-L   | 43.54 | 20.37 | 39.92 |
>
> Overall, incorporating ROUGE-L leads to the same conclusions as using only ROUGE-1 and ROUGE-2. An interesting direction for future work would be to investigate the contribution of different types of supervision signals, including BertScore and MoverScore which measure semantic relevance beyond lexical matching.

---

### Decision · Program_Chairs · 2023-01-20

**Decision:**

Accept: poster

**Justification For Why Not Higher Score:**

The contribution is focusing on extractive text summarization, which is currently "out of fashion" and it isn't clear will ever again have the prominence  of abstractive summarization work -- thus, despite being a well-motivated and well-executed paper, the impact and interest may be limited by the specific application.

**Justification For Why Not Lower Score:**

The paper is of high technical quality; well-motivated, well-written, simple, and effective. While the contribution focuses on extractive summarization, it also may impact all extractive summarization methods and may inspire work on validation methods, etc. in other text generation settings.

**Metareview: Summary, Strengths And Weaknesses:**

The authors address two observed problems in generating training sets for extractive summarization: (1) greedy/beam search in sentence selection frequently doesn't lead to an 'optimal' summary (in a structured prediction sense) and (2) greedy/beam search leads to a deterministic output that doesn't have reliable confidence calibration. To solve this, they build on recent work for soft-labels in deep learning to generate sentence-wise soft labels that can generate multiple reference summaries incorporated into extractive summarization training; referred to as ORacle ExpectatiOn (OREO) labeling. Experiments are conducted on multiple widely-used benchmark summarization datasets against multiple illustrative and competitive baselines in 'standard' supervised extractive, zero-shot cross-domain, zero-shot cross-lingual, and supervised abstraction settings -- demonstrating solid performance (often SotA with improved generalization properties).

The consensus strengths of this work include:
- They identify a systemic problem in extractive summarization and build on recent work regarding training on soft-labels in deep learning settings to develop an application-suitable soft-labeling scheme. The work is well-motivated and the approach is interesting and effective.
- The paper is well-written and well-contextualized wrt related work both conceptually and empirically.
- Both analytical and empirical results are provided to support the primary claims of the paper.
- The proposed approach ostensibly applies to any future extractive learning algorithm (that can support multi-reference) and displays promising results for abstractive cases (under specific approaches).
- The authors were able to address issues raised during rebuttal effectively; hopefully these will be integrated into the final version of the paper effectively.

The weaknesses discussed in the reviews (and my own reading) of this work include:
- Extractive summarization isn't as widely studied and specific use cases are likely to be more esoteric in the future (under current summarization use cases). Thus, the practical impact may be limited.
- The connections to offline RL are fairly strong; thus, some of the findings could be more closely related from a conceptual and analytical cases. Making these connections more rigorously would strengthen the paper.
- Soft-labeling is common in deep learning; the contributions are fairly specific to extractive summarization.

Overall, the consensus is that this is a well-executed and potentially important contribution to extractive text summarization. The proposed method (OREO) will likely improve most existing summarization techniques in terms of performance and robustness -- and may apply to other text generation problems with additional work. The primary concern is that this work centers around extractive summarization, which is less studied due to LLMs, but the strength of the contributions outweigh this limitation as it is still a widely used setting.


**Note From Pc:**

if the above contains the word "oral" or "spotlight" please see: "oral" presentation means -> notable-top-5% and "spotlight" means -> notable-top-25%. As stated in our emails, we are disassociating presentation type from AC recommendations